

# A shift in circadian stem increment patterns in a Pyrenean alpine treeline precedes spring growth after snow melting

Helen Flynn[1,2], J. Julio Camarero[2], Alba Sanmiguel-Vallelado[3], Francisco Rojas Heredia[2], Pablo
Domínguez Aguilar[2], Jesús Revuelto [2], Juan Ignacio López-Moreno[2]

[1]Department of Geosciences- Colorado State University, Fort Collins, Colorado, 80521, USA
[2]Instituto Pirenaico de Ecología (IPE-CSIC), Zaragoza, 50059, Spain
[3]iuFOR, EiFAB, Universidad de Valladolid, Campus Duques de Soria, Soria, 42004, Spain.

*Correspondence to*: Helen Flynn (Helen.Flynn@colostate.edu) or Juan Ignacio López-Moreno (nlopez@ipe.csic.es)

**Abstract.** Changing snow regimes and warmer growing seasons are some climate factors influencing productivity and growth of high-elevation forests and alpine treelines. In low-latitude mountain regions with seasonal snow and drought regimes such as the Pyrenees, these climate factors could negatively impact forest productivity. To address this issue, we assessed the relationships between climate, snow, and inter- and intra-annual radial growth and stem increment data in an alpine *Pinus uncinata* treeline ecotone located in the central Spanish Pyrenees. First, we developed tree-ring width chronologies of the study site to quantify climate-growth relationships. Second, radial growth, tree water deficit, and shrinking/swelling cycles were quantified and identified at monthly to daily scales using fine-resolution dendrometer data. These variables were extracted for three climatically different years, including one of the hottest summers on record in Spain (2022), and were related to soil water content, soil and air temperature, and the dates of snow duration across the treeline ecotone. Warmer February and May temperatures enhanced tree radial growth, probably because of an earlier snow melt-out and start of the growing season and higher growth rates in spring, respectively. The characteristic circadian cycle of stem increment, defined by night swelling and day shrinking, was detected in summer and autumn. However, this pattern inverted during the snow season from November through April, suggesting a transition phase characterized by wet soils and swollen stems preceding the spring onset of growth. Air temperature, soil temperature and moisture, and the presence of snow are strong indicators of how much and for how long mountain trees can grow. Shifts in daily stem increment patterns reveal changes in early growth phenology linked to snow melting.



## 1 Introduction

In mountain areas, warming rates are much higher than in lowlands (Pepin et al., 2015) leading to changes in seasonal snow regimes and soil moisture available for tree growth (Harpold & Molotch, 2015). Forest productivity and tree growth in high-elevation, subalpine forests and alpine treelines are especially sensitive to the warming effects of climate change (Albrich et al., 2020). However, drier conditions could also negatively impact on low- to- mid-latitude mountain forests making the late-winter soil moisture coming from snow melting critical for tree growth. This has been observed in mountain areas subjected to strong snow and drought seasonality, such as the southern Rockies, the central Andes, and often in Mediterranean mountain ranges like the Pyrenees (Andrus et al., 2018; Saavedra et al., 2018; Vicente-Serrano et al., 2021; Villalba et al., 1994). High elevation treeline forests are limited in productivity by climate variables like air and soil temperature (Peterson, 1998; Sanmiguel-Vallelado et al., 2021). Such forests are especially sensitive to the warming effects of climate change (Albrich et al., 2020) and can experience longer and, thus, more productive growing seasons (Yang et al., 2024).

Snow distribution and processes are clearly impacted by forest structures, but forests are also influenced by climate and snow as well. For instance, productivity was found to decrease with greater seasonal snow water equivalent (SWE) and later timing of melt out  in the United States and southeast Canada (Yang et al., 2024). Similarly, greater snowpacks led to less radial growth in mountain pine (*Pinus uncinata* Ram.) high-elevation forests in the Spanish Pyrenees (Sanmiguel-Vallelado et al., 2019), whereas monthly growth rates were found to be enhanced by a higher soil temperature linked to an earlier snow melt (Sanmiguel-Vallelado et al., 2021). To make compatible these different findings found at yearly to monthly scales, we need closer, finer-resolution approached to understand how climate factors, and particularly snow dynamics, drive tree radial growth, a proxy of carbon uptake in woody tissues.

Snow is an essential water resource in the Spanish Pyrenees, and is highly variable across the landscape, especially in the high-mountain treeline ecotone (Revuelto et al., 2017). Pyrenean forests have unique controls on snow distribution, redistribution, and sublimation which vary with forest density and structure (Hedstrom & Pomeroy, 1998; López-Moreno & Latron, 2008; Pomeroy et al., 2002; Revuelto et al., 2015; Storck et al., 2002). For this reason, it is important to gain a better understanding of how individual tree growing in alpine treelines respond to snowpack. Snow and vegetation dynamics are being studied more often (Dobbert et al., 2022; Huang et al., 2023; Pomeroy et al., 2006; Sanmiguel-Vallelado et al., 2019, 2021; Yang et al., 2024), specifically in treeline ecotones due to their relevance in forecasting future treeline shifts (Hagedorn et al., 2014; Huang et al., 2023).

The central Spanish Pyrenees, due to its strong seasonality in snow cover and soil water availability during the growing-season, provide an adequate setting to study snow-forest interactions. In this area, treelines are dominated by mountain pine, which shows a growing season from May through October (Camarero et al., 1998). Radial growth rates of this species peak in May



(Sanmiguel-Vallelado et al., 2021). In the Pyrenees, the areas above 1600 meters are snow covered at least 50% of the time

December through April with wet soil and low temperature conditions (Gascoin et al., 2015). However, climate change is shifting the snow regimes in the Spanish Pyrenees leading to a shorter snowpack duration due to a lower snow accumulation (López-Moreno, 2005; López-Moreno et al., 2020). Summer heatwaves and droughts are becoming common. This was illustrated by the summer of 2022 which was the hottest in the last several hundred years (Serrano-Notivoli et al., 2023); Izaguirre et al.). In addition to increasing summer temperatures, climate change is increasing vapor pressure deficit (VPD)

which could negatively impact forest productivity and growth by rising water evaporative demand (Noguera et al., 2023).

Trees swell and shrink throughout each day as a function of changes in soil moisture and atmospheric water demand usually expressed as VPD. In late winter to early spring, usually from February to April, the snow has melted out and soils are wet, but later in the growing season, higher temperatures dry soils, and increase the drought stress (Zweifel et al., 2016). During

times of high heat and low soil moisture, trees can be impacted by drought stress causing them to undergo reversible shrinking which can be quantified through a metric called tree water deficit (TWD) derived from dendrometer measurements (Zweifel et al., 2016). Trees predominantly exhibit drought stress (high TWD) during daylight hours which is why most radial growth is occurring during the night during the growing season (Zweifel et al., 2021). Tree stems are shrinking during the high-stress, TWD daylight hours, and they are expanding and contributing to irreversible radial growth during the cooler night hours. This

mechanism for growth is exhibited during the time of year when it is advantageous for the tree to do so. Circadian approaches have been used to identify this trend in other biomes with varying results (Mei-Jun et al., 2023; Ziaco & Biondi, 2018). Climatological conditions in the central Spanish Pyrenees vary greatly over the course of the year which could cause a shift in daily circadian cycles. If soil temperature and moisture, controlled by the presence of snow, drive the start of the growing season in high mountain forests and alpine treelines (Sanmiguel-Vallelado et al., 2021), then soil temperature and the presence

of snow dictate the magnitude and timing of tree growth in the treeline ecotone with the shift being dictated by a change in the daily circadian rhythm of shrinking and swelling dynamics.

The purpose of this research is to test this hypothesis and to quantify the influence of varying climate and snow conditions on radial growth in a Pyrenean treeline ecotone at several time scales (year, month, day and hour). To do this, we: 1. compared

tree growth and shrinking and swelling dynamics across three climatically differentiated years, 2. analyzed daily stem radius fluctuations as related to changes in growth and shrinking/swelling dynamics during warm and cold seasons, and 3. evaluate how climatic variables (air/soil temperature, snow, precipitation, soil water content) affect growth and TWD.



## 2 Materials and Methods

### 2.1 Study Site

The study site is a relatively undisturbed treeline ecotone located in the Sierra de las Cutas (42.6371 ºN, 0.0512º W), near the "Ordesa and Monte Perdido" National Park in northeastern Spain. The site sits at an average elevation of 2100 meters facing 186º south-southwest and is dominated by mountain pine. The average slope is 17º, reaching a maximum slope of 33º when descending towards the forest. Soils are rocky and basic. The understory is dominated by several shrub species such as *Juniperus communis* L. above the treeline and *Rhododendron ferrugineum* L. and *Calluna vulgaris* L. within the forest (Pardo

et al., 2013).

### 2.2 Climate and dendrometer data

In an open area in the upper limit of the tree line, the automatic weather station (AWS) at Las Cutas is equipped to measure air temperature, relative humidity, wind velocity, and incoming solar radiation. The data from 2020 to 2024 were downloaded and processed which revealed a gap in sensor data from December 6, 2021, to June 17, 2022 due to battery malfunction (Fig.

1). There is a meteorological station 2 km away at the Góriz refuge (42.6634 ºN, 0.0148º E) with similar sensors, an additional precipitation gauge and a snow depth sensor. In addition to the AWS at Las Cutas, there were also four soil water content (SWC) sensors (ECH2O probe, model EC-5, Decagon Devices, Pullman, Washington, USA) installed at 10cm depths in the forested zone of the tree line ecotone. One of the four sensors malfunctioned, so SWC data was averaged across the three remaining sensors (Fig. 1). Other stand-scale climatological data were downloaded from the nearby Góriz refuge. The refuge

lies on a similarly south-facing aspect at an elevation of 2200 m. The Góriz meteorological station provided additional daily air temperature, precipitation and snow depth data. After correlating the existing daily minimum and maximum air temperature data from Las Cutas and the Góriz refuge, it was determined that the Góriz data could supplement the data from Las Cutas (Fig. A1).





**Figure 1. Daily mean air (T$_{air}$) and soil (T$_{soil}$) temperatures, and mean soil water content (SWC) in the three study years. Values are means ± SE. Snow cover dates are outlined in vertical blue boxes with inconsistent snow cover defined by light blue and consistent snow cover defined by dark blue.**

Soil temperature at each tree was recorded each hour using a combination of dataloggers (Tinytag-Plus-2, model TGP-4017, Gemini Dataloggggers UK Ltd., Chichester, West Sussex, UK; EasyLog-USB, model EL-USB-1 PRO, Lascar Electronics Ltd., Whiteparish, Wiltshire, UK; Thermochron iButton, model DS-1922L, Dallas Semiconductors, Texas, USA). Data were downloaded and sensors replaced approximately every 6 months. Due to sensor malfunction, only 6 of the 9 trees had a complete T$_{soil}$ time series for the period of record. Extraneous T$_{soil}$ values (greater than 100ºC away from 0ºC) were removed.

In September 2020, stainless steel band dendrometers (DR 26, EMS Brno, Czech Republic) were placed on 9 *P. uncinata* individuals in three zones of the tree line ecotone: the forest (TRA), transition to tree line (TRA-TRE), and treeline (TRE).



The dendrometers recorded 15-minute perimeter measurements at a high resolution of 1 µm and air temperature ($T_{air}$) until October 2024. The diameter at breast height (DBH) and height were also measured for each individual tree (Table A1) when the dendrometers were installed in 2020. Radius values calculated using the perimeter measurements (assuming a spherical

circumference) were then processed and cleaned using the program treenetproc in R (Knüsel et al., 2021). To examine the daily circadian rhythms of the trees, hourly stem radial increment values were calculated using the maximum hourly radius, and then normalized across the 9 individuals. Hourly stem radius values were then averaged by month across all years to view the changes in daily shrinking and swelling patterns over the time period by month.

### 2.3 Tree-ring width data and processing

In late October 2023, 20 mature trees were selected and 2 cores per tree were taken at 1.3 m using a Pressler increment borer. Cores were air dried in the laboratory, glued to wooden mounts, and sanded with progressively finer sandpapers for visualizing tree-ring boundaries (Fritz, Harold C, 1976). Then, they were scanned at 2400 dpi resolution and visually cross-dated under a stereoscope. Ring widths were measured with a 0.001 mm resolution using scanned images and the CooRecorder-CDendro software (Maxwell & Larsson, 2021). The quality of cross-dating was checked using the COFECHA software which calculates

moving correlations between individual series of ring-width values and the mean sites series (Holmes, 1983).

To calculate climate-growth relationships, first the individual ring-width series were detrended by fitting x-year cubic smoothing splines with a 50 % frequency response cut-off, where x was 2/3 of the mean series length. The measured ring-width values were divided by fitted values. The resulting dimensionless ring-width indices were pre-whitened by fitting auto-

regressive models and were averaged using bi-weight robust means. This allowed building a mean series or chronology of ring-width indices preserving annual to decadal variability. Lastly, calculated over the common 1970–2023 period, an Expressed Population Signal (EPS) ≥ 0.85 indicated a high common signal of the chronology (Wigley et al., 1984). Lastly, Pearson correlations were calculated from the prior September to the current September between climate data and the ring-width indices. Monthly climate data (Tx, mean maximum temperature; Tn, mean minimum temperature; Pr, precipitation)

were obtained from the 0.1°-gridded E-OBS dataset v29.0e (Cornes et al., 2018) for the period 1970–2023. These procedures were done using the dplR ( Bunn et al., 2023; Bunn, 2008, 2010) and treeclim (Zang & Biondi, 2015) R packages.

To verify that the patterns identified in those studies were consistent with the Las Cutas experimental site, monthly correlations between climate variables (Tx, mean maximum temperature; Tn, mean minimum temperature; Pr, total precipitation) and the

mean series of ring-width indices (period 1970-2023) were calculated using the Pearson method for the September prior to the growing season of interest and up until the September after the growing season of interest. The results of that analysis were used to identify which climate variables during which months showed the highest correlation to growth. Using that information, climate summaries from the months deemed influential were compared across years to identify possible interannual variability and test the hypothesis.





### 2.4 Relating climate and dendrometer data


Daily stem radial increment time series were calculated from the sub-hourly dendrometer data using the maximum daily radius following a daily approach (Deslauriers et al., 2007). The growing season was defined as day of year (DOY) 100 through 300 (Camarero et al., 1998; Sanmiguel-Vallelado et al., 2021). Using the daily radial increment data, the daily growth rate and TWD were extracted from the time series (Zweifel et al., 2016). The correlation between daily $T_{air}$ and the growth rate was calculated during the growing period for values from the same day, as well as a lagged correlation between the growth rate and $T_{air}$ from the previous day up to 10 days prior using the Pearson method. This correlation was also calculated between the SWC and TWD. In addition, 20-day moving correlations were calculated for daily $T_{air}$ and the growth rate, daily $T_{soil}$ and growth rate, and SWC and TWD.



With the complete $T_{soil}$ data, daily temperature oscillation (DTO) was calculated by subtracting the daily minimum temperature from the daily maximum temperature. To determine the time periods in which snow was present, days with snow cover were identified as having a maximum DTO (DTOmax) of less than 2 degrees and less than 5 degrees. These two datasets for each of the 6 trees with $T_{soil}$ data were used to define the snow cover duration periods. Snow cover was considered intermittent starting (IS) on the first date with snow presence and before the continuous (CS) period which was defined as the period with gaps less than 1 day. When gaps larger than 1 day began to occur again in the spring, the inconsistent period resumed until gaps with snow cover exceeded 15 days.


Because of the similar location, climatological conditions, and strong correlation between the $T_{air}$ recorded at the Góriz refuge and the AWS at Las Cutas (Fig. A1), the calculated snow cover duration dates were compared with the Góriz refuge snow depth data. Although the refuge sits at an elevation approximately 100 meters higher than the Las Cutas average, the dates corresponded to accumulation and melting patterns identified in the Góriz snow depth data. The snow cover duration dates with a DTOmax less than 2 were determined to be more accurate and were used for the rest of the analysis.


### 3 Results

Each year of the study period showed varying tree growth and shrinking and swelling dynamics. In 2021, the onset of growth and day of maximum growth rate were slightly delayed compared to the other two years in which the maximum growth rate occurred 10 days earlier in 2022 and 15 days earlier in 2023, on average (Table 2). In addition, the length of the 2021 growing season was the shortest at an average of 83 days while the 2022 season was the longest with an average of 144 days (Table 2).





**Table 2. Summary statistics of each growing season. All values are averages ± SE. Different letters indicate significant (p < 0.05) differences between years according to *t* tests.**

| Variables | Year | | |
|---|---|---|---|
| | 2021 | 2022 | 2023 |
| Growth start (DOY) | 151 ± 3a | 143 ± 2b | 136 ± 2c |
| Growth end (DOY) | 249 ± 2a | 287 ± 5b | 244 ± 2a |
| Growth length (days) | 83 ± 4a | 144 ± 5b | 108 ± 3c |
| Maximum growth rate (mm day$^{-1}$) | 19.7 ± 3.1 | 24.7 ± 2.4 | 22.7 ± 2.1 |
| Date of maximum growth rate (DOY) | 167 ± 2a | 157 ± 3b | 152 ± 3b |

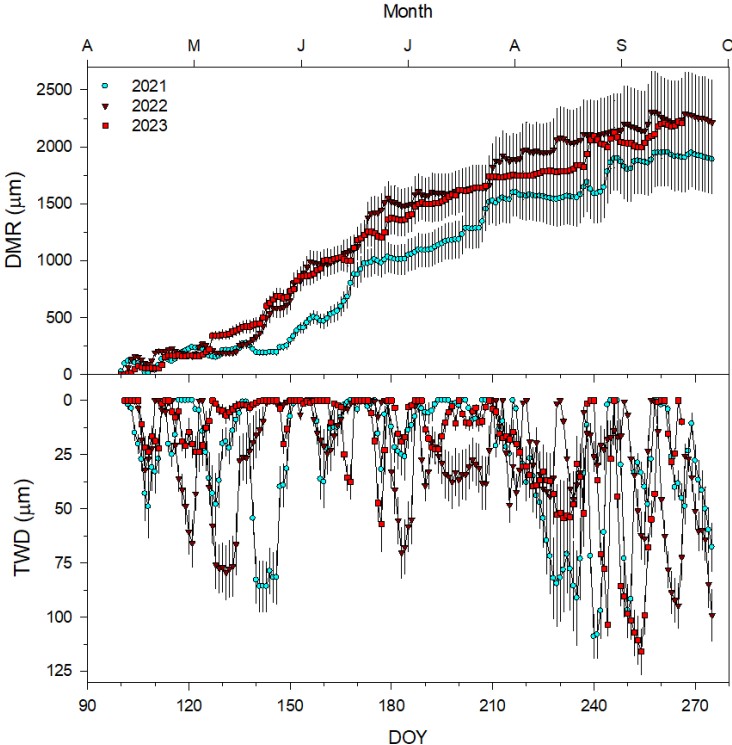


**Figure 2. Daily values (means ± SE) of radial stem variation (DMR) and minimum tree water deficit (TWD) in the three study years. DOY is the day of calendar year.**



The average DMR in 2022 and 2023 showed increased growth rates starting around DOY 130. The 2022 growing season was the most gainful (approximately 2300 µm) of the three examined, while the 2021 season showed the least total growth

(approximately 1900 µm) (Fig. 2). The TWD also varied in timing between years. During the 2022 growing season, there were two large (greater than 60 µm) peaks before DOY 140 (Fig. 2). There was also one large peak in TWD in 2021 around DOY 140 and DOY 225 (Fig. 2).

TWD had high intra- and interannual variability throughout the time series (Fig. 2). There were small peaks in TWD (less than

100 µm) around DOY 145 and 130 in 2021 and 2022 respectively (Fig. 2). However, the largest peaks occurred in the late growing season between DOY 225 and 280 with 2023 demonstrating the largest peak (Fig. 2).

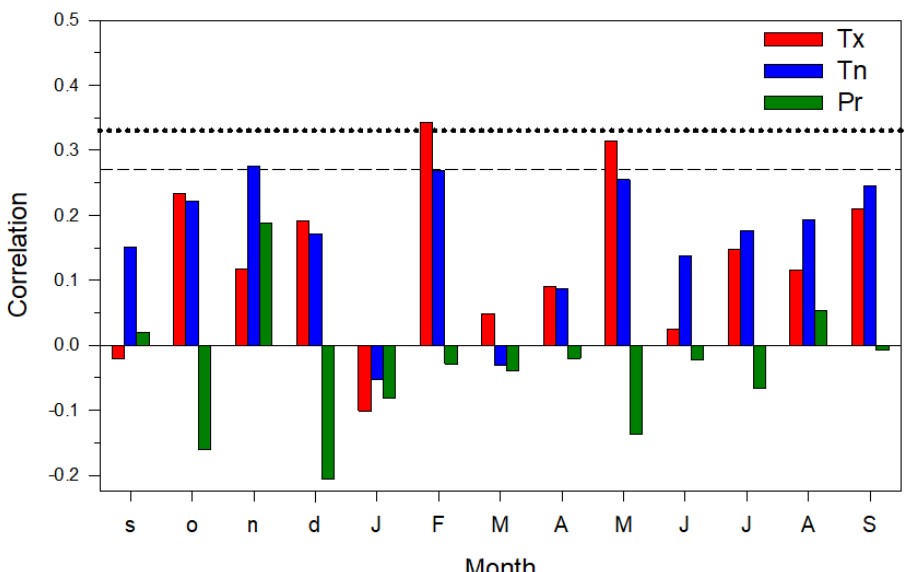

**Figure 3. Climate-growth relationships (Pearson correlations) calculated by relating monthly climate variables (Tx, mean maximum**

**temperature; Tn, mean minimum temperature; Pr, total precipitation) and the mean series of ring-width indices (period 1970–2023). Correlations were calculated from September of the previous year (months abbreviated by lowercase letters) to September of the current year (months abbreviated by uppercase letters). Dashed and dotted horizontal lines show the 0.05 and 0.01 significance levels, respectively.**

Using the results of the ring-width and climate correlation, mean maximum air temperature and mean minimum air temperature

from the months of February and May showed the highest correlation to growth (Fig. 3). Thus, climate summaries were compared across years for these months and the overall growing season (Table 3).



**Table 3. A summary of the climate variables at the study site and precipitation from the nearby Góriz station. Values are means ± SE.**

| Climate variable | Time period | 2021 | 2022 | 2023 |
|---|---|---|---|---|
| **Soil temperature (ºC)** | February | 0.73 ± 0.45 | 1.72 ± 0.55 | 0.84 ± 0.24 |
| | May | 5.88 ± 0.27 | 8.77 ± 0.79 | 7.12 ± 0.57 |
| | Growing season | 9.89 ± 0.52 | 10.54 ± 0.86 | 11.45 ± 0.66 |
| **Air temperature (ºC)** | February | 1.14 ± 0.47 | 2.5 ± 0.57 | 0.06 ± 0.82 |
| | May | 6.22 ± 0.63 | 10.41 ± 0.82 | 6.7 ± 0.6 |
| | Growing season | 10.00 ± 0.37 | 12.39 ± 0.37 | 12.04 ± 0.39 |
| **SWC ($m^3*m^{-3}$)** | February | 0.20 ± 0.01 | —— | 0.25 ± 0.01 |
| | May | 0.18 ± 0.01 | ——— | 0.21 ± 0.01 |
| | Growing season | 0.15 ± 0.01 | 0.15 ± 0 | 0.18 ± 0.01 |
| **Precipitation (mm)** | February | 1102 | 51 | 79 |
| | May | 1044 | 22 | 118 |
| | Growing season | 6844 | 945 | 114 |

The average air temperatures in February, May and the fixed growing season (DOY 100-300) were highest in 2022, then 2023 (Table 3). For instance, in 2011, 2022 and 2023, growing-season $T_{air}$ were 10.0, 12.4 and 12.0 ºC, respectively. This was also

true of the average soil temperatures except for the fixed growing season in 2023 which had average soil temperatures slightly less than 1 ºC higher than in 2022 (Table 3). For instance, in 2021, 2022 and 2023, growing-season $T_{soil}$ were 9.9, 10.5 and 11.4 ºC, respectively. Due to SWC sensor malfunction in 2022, data are limited to the fixed growing season. The average SWC was slightly higher in February, May and the fixed growing season of 2023 than 2021 (Table 3).

**Table 4. Calculated snow season dates are included with IS meaning intermittent snow season and CS meaning continuous snow**
**season.**

| Snow season index | 2021 | 2022 | 2023 |
|---|---|---|---|
| $IS_{start}$ (DOY) | 275 ± 1 | 320 ± 4 | 292 ± 1 |
| $CS_{start}$ (DOY) | 331 ± 4 | 339 ± 8 | 334 ± 7 |
| $CS_{end}$ (DOY) | 87 ± 5 | 117 ± 49 | 22 ± 15 |
| $IS_{end}$ (DOY) | 128 ± 11 | 118 ± 1 | 84 ± 5 |
| length of CS (days) | 121 ± 9 | 113 ± 26 | 64 ± 15 |
| length of IS (days) | 218 ± 11 | 163 ± 3 | 157 ± 5 |
| days w/ snow during IS (days) | 156 ± 13 | 119 ± 13 | 106 ± 11 |
| % IS with snow | 0.72 ± 0.06 | 0.73 ± 0.08 | 0.68 ± 0 |



Precipitation from the Góriz refuge also varied highly between the first two years of this study. In 2021, the cumulative precipitation in February was 1102 mm (Table 3). This dropped to 51 mm in 2022, then rose slightly in 2023 for a total of 79 mm (Table 3). Similar trends were observed in May as well (Table 3). However, the total precipitation during the entirety of

each growing season was 684 mm, 945 mm, and 114 mm for the years 2021, 2022, and 2023, respectively (Table 3).

Although snow presence was not directly recorded, days with snow were estimated using soil temperature. The longest consistent snow season occurred during the 2020-2021 winter, followed by 2021-2022, and, finally, 2022-2023 (Table 4). Unlike the two other years which appeared to have a longer period of inconsistent snow towards the end of the winter, the 2021-2022 winter snow cover ended abruptly around DOY 118 ± 1, which was earlier than the 2020-2021 winter (DOY 128

± 11) and later than the 2022-2023 winter (DOY 84 ± 5) (Table 4).

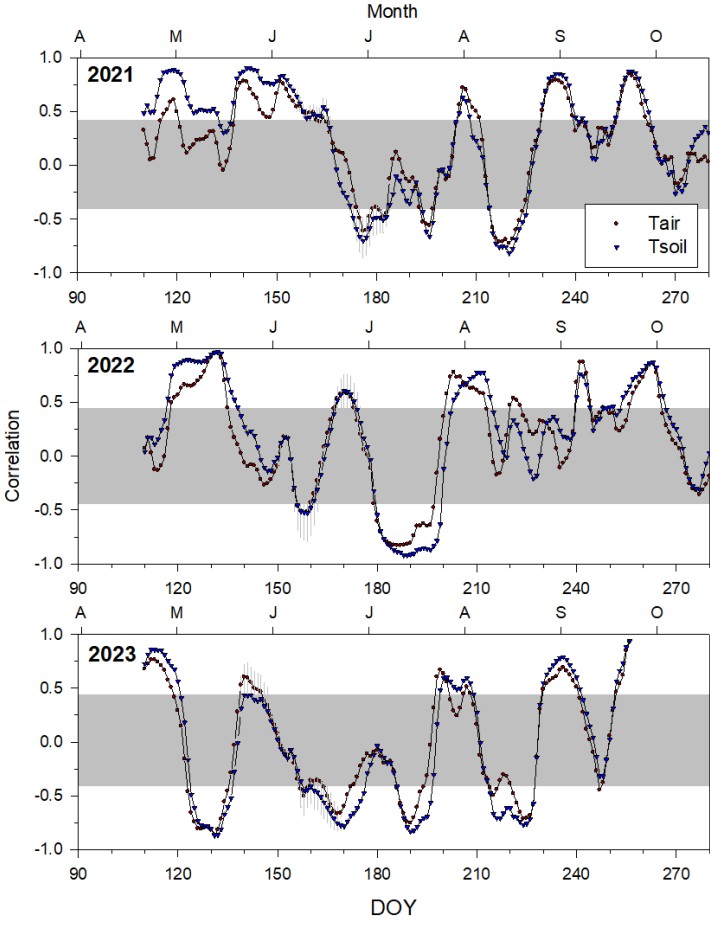

**Figure 4. Moving correlations calculated by correlating air ($T_{air}$) or soil ($T_{soil}$) temperatures with daily growth rates along 20-day periods. Values are means ± SE. Correlation values located outside the grey box are significant ($p < 0.05$).**

The moving correlations between $T_{air}$ or $T_{soil}$ with the growth rate showed generally similar patterns for the fixed growing

seasons during the years 2021 and 2022 (Fig. 4). Both years had a window with a peak in positive significance for both $T_{air}$



and $T_{soil}$ during May (Fig. 4). This pattern appears to have occurred earlier in 2023 with the peak in significance already beginning to decline by the start of May. The year 2023 also stands out as the year with the smallest difference between $T_{air}$ and $T_{soil}$.

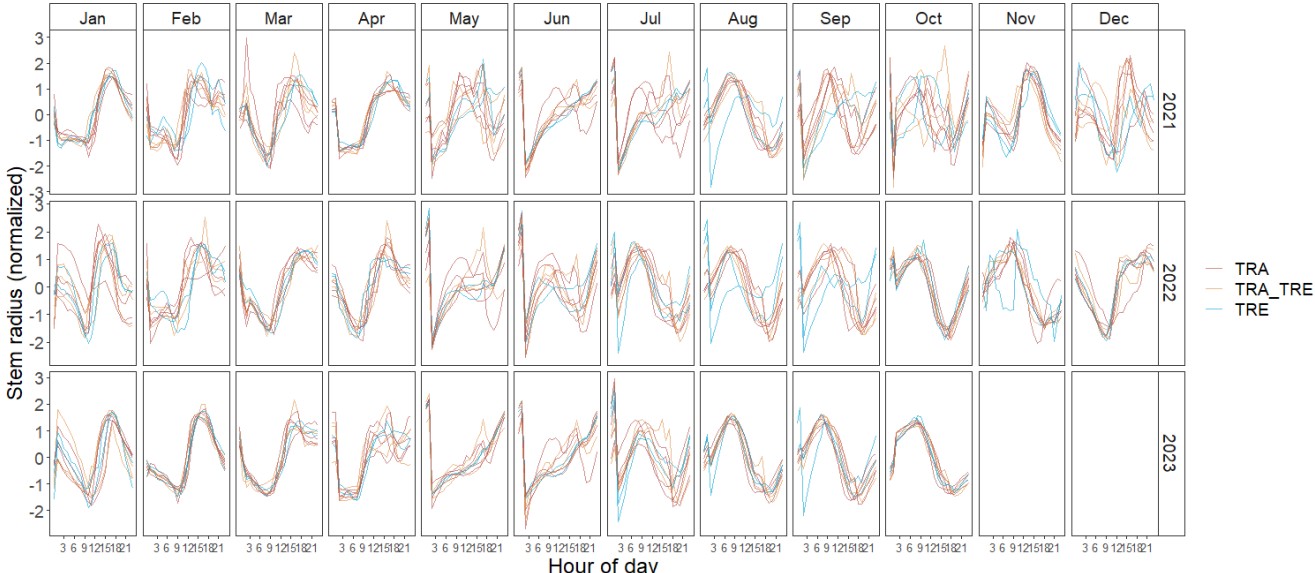

**Figure 5. The hourly stem radius normalized between trees and averaged by month with each solid line representing an individual tree in the forest (TRA), transition to treeline (TRA-TRE) and treeline (TRE) zones.**

The in-depth analysis of the seasonal changes in stem shrinking and swelling showed that the daily pattern shifts across the seasons (Fig. 5). The growing season pattern shows swelling during the evening and shrinking during the day beginning in July or August and ending in November (Fig. 5). However, the pattern that can be seen during the snow season (November or December through April) is inverted, meaning that it is often the reverse of the normal growing season daily pattern exhibiting expansion during the day and shrinking during the night (Fig. 5). The switch from the inverted pattern to the normal pattern occurred very rapidly during the months of May or June, as noted by the decoupling of the cycle between trees (Fig. 5).

## 4 Discussion and Conclusions

The onset of the growing season at this site is triggered by rising air and soil temperatures, and the disappearance of the winter snowpack which is consistent with previous studies conducted on the study species (Galván et al., 2014; Tardif et al., 2003). While important, May air and soil temperatures were less significant when correlated to total growth than February air and soil temperatures. In winters like 2022 that are warmer in February, there is likely less precipitation falling as snow and accumulating which leads to an earlier melt out date, earlier warming of soil, and longer growing seasons resulting in more total growth. In a previous study, February was identified as an important month in the climate-growth relationships



specifically between air temperature and radial growth (Sanmiguel-Vallelado et al., 2019). Additionally, researchers found that radial growth in *P. uncinata* was related to early and late growing season soil temperatures (Sanmiguel-Vallelado et al., 2021). They specifically found that May climatological conditions were more strongly correlated with growth than other months (Sanmiguel-Vallelado et al., 2021).

The high interannual variability of this region allows for the comparison of the impact of climatological differences on *P. uncinata* growth and TWD. Higher February air and soil temperatures in 2023 and especially in 2022 lead to an earlier growth onset, and a longer growing season which is consistent with the findings in (Sanmiguel-Vallelado et al., 2019). In summer, the central Spanish Pyrenees become hotter and drier leading to high vapor pressure deficit (VPD) and increased TWD. Under this set of environmental conditions, trees tend to close its stomata to conserve water during the hottest hours of the day

(Grossiord et al., 2020). For this reason, it is common for trees to grow at night during months when VPD and TWD during the day is too high (Tumajer et al., 2022; Zweifel et al., 2016). The normal growing season pattern was identified in previous research (Zweifel et al., 2016) and occurs during the warm months at Las Cutas, beginning in July or August and ending in October or November. This pattern could be attributed to normal responses to high VPD. During the cold months, when trees were not experiencing high VPD, and had much lower daily growth rates, radial increment swelling occurred during daylight

hours. This switch in the swelling/shrinking cycle was attributed to the presence of snow which was influenced by winter air temperatures and led to cold and moist soil conditions.

In agreement with prior studies, we observed that warmer February temperatures, specifically soil temperatures, enhanced overall growth and increased spring growth rates, and this positive effect was related to an earlier snow melt-out and, thus, a

longer growing season. The characteristic circadian cycle of stem increment that leads to wood production, defined by night swelling and day shrinking, was detected in summer and autumn (approximately from July to November) during the growing season. However, this pattern was inverted during the snow season (approximately from November to April), prior to the onset of growth (Fig. 6). Air temperature, soil temperature, and the presence of snow are strong indicators of how much and for how long trees can grow each year and cause the shift in daily stem increment patterns of tree radius.




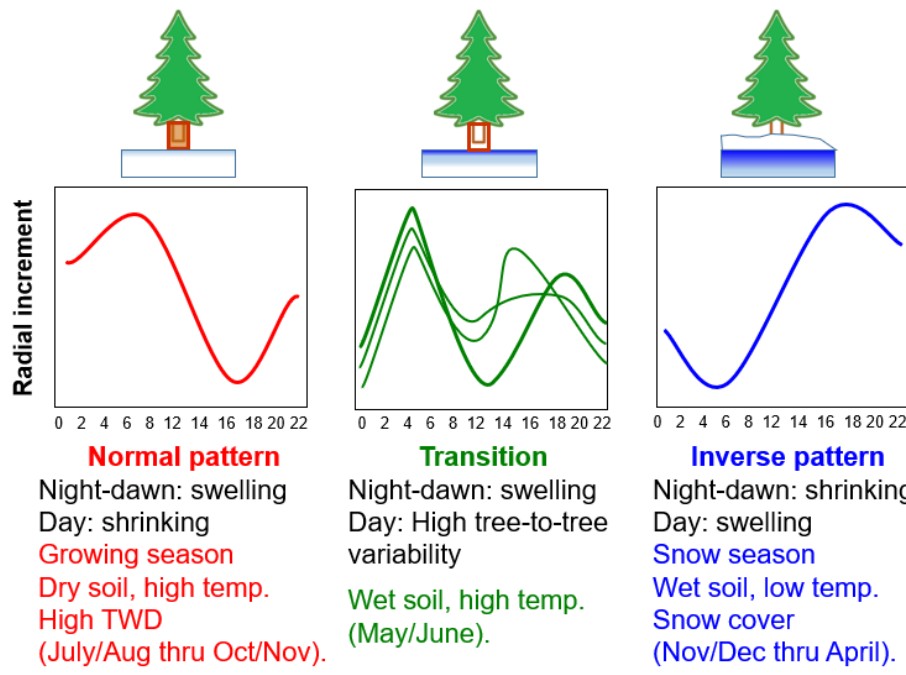

**Figure 6. Graphic summarizing the patterns found in this study and showing the shifts in snow and climate conditions and swelling/shrinking stem cycles along the growing season.**


Future research could use in-situ and short-range remotely sensed snow observations to confirm that snow is the driver of the changes to the growing season and total growth that we saw in this study. Different locations of each tree (within the stand, outer edge, or krummholz) could also impact the timing of snow melt, the growth rate, total growth, TWD and circadian stem increment cycles. In addition, dates of growth onset are inferred from dendrometer data, but they should be confirmed using

periodic wood monitoring and xylogenesis analyses (e.g. Sanmiguel-Vallelado et al., 2021). Unfortunately, we were limited by the number of sample trees in each zone to make any conclusions, so this would be a good starting point for another study.

**Author contributions:**

Conceptualization, J.J.C., A.S.V., H.F., J.I.L.M.; data curation and management, H.F., J.J.C., A.S.V., F.R.H.; formal analysis, H.F., J.J.C., A.S.V.; methodology, A.S.V., J.J.C, H.F., P.D.A.; resources, J.I.L.M., J.J.C., J.R.; writing - original draft

preparation, H.F.; writing - review and editing, H.F., A.S.V., J.J.C, J.I.L.M., J.R., F.R.H., P.D.A.; visualization, J.J.C., H.F., A.S.V.; project administration, J.I.L.M., J.J.C., J.R.



**Funding:**

This research was funded by a Fulbright Spain Predoctoral Fellowship to H.F. A.S.V. was supported by postdoctoral grant

JDC2022-048316-I funded by MICIU/AEI/10.13039/501100011033 and by European Union NextGenerationEU/PRTR.

**Code Availability:**

The code used in the analysis of this research is available upon request from the first author.

**Data Availability:**

The data used in this research are available upon request from the first author.

**Competing interests:**

The authors declare that they have no conflicts of interest.

**Acknowledgements:**

The authors would like to thank the members of CryoPyr for carrying out the fieldwork, the Agencia Estatal de Meteorología

of Spain for providing the Góriz refuge data, and the Pyrenees Institute of Ecology – Zaragoza for hosting H.F.

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



**Appendix:**

**Table A1. Tree features (dbh, height, location).**

| Tree Nº | DBH (cm) | Height (m) | Location | Location |
|---------|----------|------------|----------|----------|
| 311 | 18.9 | 14.6 | TRA | Forest |
| 326 | 19.0 | 15.0 | TRA | Forest |
| 306 | 14.0 | 13.2 | TRA | Forest |
| 316 | 18.4 | 14.0 | TRA | Forest |
| 314 | 14.7 | 11.0 | TRA-TRE | Transition |
| 304 | 12.2 | 8.5 | TRA-TRE | Transition |
| 798 | 12.7 | 9.0 | TRA-TRE | Transition |
| 799 | 20.0 | 6.0 | TRE | Treeline |
| 800 | 8.6 | 2.8 | TRE | Treeline |





**Figure A1. a) Maximum and b) minimum daily temperatures at Las Cutas (treeline site) and the Góriz refuge (local station).**

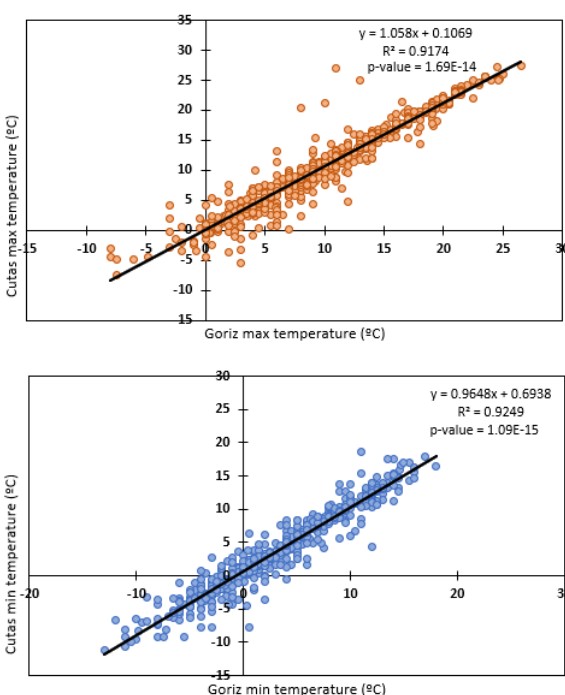