# Peer review of "A shift in circadian stem increment patterns in a Pyrenean alpine treeline precedes spring growth after snow melting"

_EGUsphere, 2024_

## Author Response (AR1)

Wisconsin, 24/12/2024

Dear editor,

We are pleased to submit a revised version of our manuscript entitled: "A shift in circadian stem increment patterns in a Pyrenean alpine treeline precedes spring growth after snow melting". We want to thank the two reviewers for their positive feedback and suggestions that helped to improve the manuscript and the clarity to present the main results. Below, you can find a point by point answer (A) to each question (Q) made by the reviewers.

Looking forward to hear your kind reply,

Helen Flynn and coauthors

**Reviewer 1:** Edurne Martinez Del Castillo

The manuscript egusphere-2024-3385, entitled "A shift in circadian stem increment patterns in a Pyrenean alpine treeline precedes spring growth after snow melting" tackles a timely and important ecological issue, namely the effects of climate change on high-elevation ecosystems. The authors use dendrometer data, tree-ring chronologies, and climate data to examine the influence of air and soil temperatures, snowpack duration, and soil water content on tree growth at inter- and intra-annual scales. The results highlight that warmer February and May temperatures promote earlier snowmelt and longer growing seasons, affecting the growth patterns. The figures are well designed and informative, complementing the text and effectively communicating the results. The analyses are succinct to only three years of data, but they are climatically distinct years, which adds variability to the findings. The study introduces a novel perspective on circadian stem increment cycles, giving insights into the inversion of these patterns during snow seasons and their potential phenological implications. Overall, the manuscript would be a valuable contribution to Biogeosciences, but several aspects need to be improved, see my detailed comments below.

Q1: **Introduction**

The introduction is comprehensive, but it could better synthesize the relevance of circadian cycles in tree physiology and tree line dynamics Additional studies could strengthen the literature review and help linking the ecological context of the article. Some examples:

- Lázaro-Gimeno, C. Ferrari, N. Delhomme, M. Johansson, J. Sjölander, R. Kumar Singh, M. Mutwil, M. E Eriksson (2024) The circadian clock participates in seasonal growth in Norway spruce (Picea abies), Tree Physiology, Volume 44, Issue 11. https://doi.org/10.1093/treephys/tpae139

- Lüttge, U., Hertel, B. Diurnal and annual rhythms in trees. Trees 23, 683–700 (2009). https://doi.org/10.1007/s00468-009-0324-1

L88. The hypothesis should be formulated at the end of the last paragraph to be quickly identified.

A1: We expanded the literature review using the citations you suggested, and restructured the end of the introduction to include the hypothesis at the end.

Q2. Materials and Methods

The methodological approach is robust but could be more transparent in addressing potential biases or limitations, such as the reliance on snow presence inferred from soil temperature oscillations (L170-176), which might benefit from validation using direct snowpack observations or satellite images. If this method is commonly used to calculate the period of snow presence, other studies should be cited. It is not clear to me whether snow season definitions based on temperature oscillations are an innovative approach, but they could be better validated with alternative snow metrics.

Some methodological details are sparse. For instance, while dendrometer calibration is mentioned, specific steps to address potential biases in measurements (e.g., sensitivity to environmental factors) could be elaborated. The relatively small sample size (only nine trees for dendrometer data) limits the generalizability of the conclusions.

Statistical approaches (e.g. Pearson correlations and moving averages) are valid but might benefit from additional justification regarding their selection.

A2: We agree that a justification and explanation of the snow presence data should be included. We clarified the implementation of the method. Limitations of this method and the tree sample size have been added to the discussion. The statistical methods were chosen based on their simplicity and robustness. This has been expanded upon in the text.

Result

Q3. The findings related to soil water content (SWC) are briefly discussed. Since SWC is a critical factor influencing tree growth, a more detailed exploration of its role during critical phenological transitions could enhance the discussion.

A3: We have expanded briefly on the results of the SWC data, and included a more in-depth discussion in the discussion section.

Q4: Table 2. It is not clear what the letters after the average±SE mean. The caption says that they indicate significance, but it is not clear what the difference between a, b, or c is.

A4: The caption of Table 2 has been adjusted to read. The different letters indicate significant (p < 0.05) differences between years.

Q5:Table 3. Check the precipitation values for year 2021, those values cannot be mm.

A5: Thank you for your attention to the details on Table 3. All precipitation values were reexamined and corrected.

Table 3. The total precipitation of the growing season in 2023 cannot be 114mm if there was 118 mm during May. Check the values, please.

Q6: Figure 4. I understand that this analysis starts in mid-April as this is the beginning of the radial increase of the trees, however, the highest historical correlations with climatic variables (in fact, with temperature) were detected in February. In my opinion, these climate-growth relationships are not sufficiently explored and discussed in the article.

A6: With regards to Figure 4, we discussed an analysis like you have suggested that includes the moving correlation across the entire calendar year. However, because the magnitude of shrinking and swelling is so small, the results are misleading. Additionally, we found that February and May air and soil temperatures are correlated to growing season dynamics like total growth. Thus, applying a moving correlation to February temperatures and swelling does not demonstrate the relation between February temperatures and growing season growth. I hope this clarifies our choice to only include the growing season in Figure 4.

Q7: Figure 6. This is an excellent summary figure (could be a very illustrative graphical abstract) but might benefit from additional labeling or annotation for clarity. What is the orange square on the tree trunk of the "normal pattern" and "Transition" trees? Why is one filled and the other is not?

A7: Regarding Figure 6, the rectangle around the tree trunk is meant to highlight when the trees are in an active growing phase. This has been clarified in the figure caption.

Q8 Discussion

This is the major weakness of the article, in my opinion. The discussion is rather superficial, and there is a limited exploration of the broader ecological and global implications of the findings. It effectively connects findings to prior research, but a deeper exploration of how these results might generalize to other alpine ecosystems would increase the manuscript's broader applicability. The authors could explore deeper into the ecological significance of phenological shifts, particularly their long-term impacts on carbon sequestration and forest dynamics under climate change.

While the study provides detailed and novel insights into circadian stem increment cycles and the influence of snow dynamics on tree phenology, it largely focuses on a specific alpine tree line in the Spanish Pyrenees. The findings are not sufficiently contextualized within a broader ecological framework, such as global alpine ecosystems or potential feedback mechanisms with climate change.

Additionally, the climate-growth relationships shown in the results are not sufficiently explored and are contextualized only using two research papers (Sanmiguel-Vallelado et al., 2019, 2021). For instance, in L283 the authors mention prior studies without citation.

A paragraph explaining the study's limitations is needed. For example, the small sample size of the dendrometer data (i.e., nine trees) undermines the generalizability of the results, especially considering the spatial heterogeneity often present in tree ecotones.

A8:Thanks for the comment. We have reworked the discussion based on your suggestions. A more detailed discussion of the SWC was included, as well as a paragraph about the limitations of the study. More citations were included to show the relation between our work and previous studies where applicable.

The latter half of this section was also restructured to include recommendations for future work and the final concluding statements at the very end.

Q9 Conclusion

L300 – The last phrase should be an overall conclusion of your findings, not a justification for further studies.

A9 Thanks again. According the restructuration of the section, the last sentence shows the main findings of the study.

**Reviewer 2:** Anonymous

The preprint provides insights into the stem circadian phenology of *Pinus uncinata* at the uppermost forest limit in the Pyrenees using high-resolution dendrometer data collected from 2021 to 2023. The manuscript is well-written and exhibits a smooth flow. The main issue concerns the limited number of trees sampled, which are characterized by varying structural parameters (ranging from seedlings to small trees) and collected from different locations (forest, transition to treeline, and treeline). However, I find the results interesting and, as mentioned by the authors, useful for guiding future research. Below comments and suggestions that could help to improve it.

Q1;Ll 119-123: "Soil temperature at each tree …", since the number of trees sampled and their locations are not specified at this point in the manuscript, I would move this paragraph after the one between lines 125 and 133.

A1: The paragraph at L119-123 was moved to the recommended location after the paragraph about the dendrometer data.

Q2 l 150: Why didn't you examine the entire period from 1950 to 2023?

A2: L150: We think including the 1950-1970 would not change the main results since beginning in 1970 gives us the 50 years prior to the first year in this study and this period includes most of study trees recruited in the past 50-60 years.

Q3: l152: This sentence is unclear; do you mean past studies?
A3: L152: This was referring to the methods mentioned in the previous 5 lines. It has been clarified in the text.

Q4Chapter 2.4: Radial growth rate is age dependent, as the radial increment decreases with increasing tree diameter. The sampled pines have diameters and heights that are not always similar, as shown in Table A1. Therefore, you should consider converting the dendrometer measurements into basal area increments.
A4: We acknowledge that there could be different physiological responses and mechanisms in the older stand that was used to create the master chronology. A mention of this has been included in the limitations section of the revised paper. Nevertheless, the tree-ring and dendrometer data refer to trees of similar age classes, namely those recruited in the past 50-60 years. In other words, dendrometers were not placed in relatively young trees. Note also that this is an alpine treeline where radial growth mainly depends on temperature variability and scarcely responds to precipitation variability.

Q5 Table 2: Maximum growth rate (µm day-1).

A6: Thanks! corrected

Q7: l 210: I would add a table to report the main statistics of the chronology built (period, mean radial growth, mean sensitivity/Gini, Rbar, etc.).

A7: We feel that the statistics generated for Table 2 are sufficient for the scope of the study and analysis that was conducted, which focused on dendrometer data. Nevertheless, these are some basic statistics characterizing the tree-ring width chronology which we added after l. 150: (values are means ± SD) ring-width = 1.65 ± 0.46 mm, first-order autocorrelation = 0.71 ± 0.17 mm, mean sensitivity of standardized width indices = 0.24 ± 0.05 mm, rbar = 0.37; EPS = 0.89.

Ll 214-216: The climate-growth relationship has been conducted on mature trees, while I assume that the dendrometer data refer to younger pines. This passage assumes that the limiting climatic factors for growth are the same in young and mature trees. I would expect, for example, that young pines (with a limited root system and growing on shallower soils at the treeline) are more sensitive to precipitation compared to those used to construct the master chronology. This could be a point to explore further in the discussions section.

Q8 Table 4: Is it possible to also add the maximum snow depth?

A8: Table 4: Maximum snow depth could not be directly measured at Las Cutas. Thus, we would need to use the Góriz refuge snow depths. Although the Góriz refuge snow data were used to confirm our calculated snow presence dates, we feel that the snow depth measurements do not exactly represent the spatial variability of the study site and could be misleading to include in Table 4.

Finally, regarding the concern over the sample size, we acknowledge that there were only 9 trees in the study, but the detailed monitoring of their microenvironmental conditions restricted the number of monitored trees.

Thank you for your comments and suggestions. We've incorporated what we feel enhances the work.